# VQ-Map: Bird's-Eye-View Map Layout Estimation in Tokenized Discrete Space via Vector Quantization

**Yiwei Zhang**[1,2]**, Jin Gao**[1,2*]**, Fudong Ge**[1,2]**, Guan Luo**[1,2]**, Bing Li**[1,2,5]**,**
**Zhaoxiang Zhang**[1,2,4]**, Haibin Ling**[6]**, Weiming Hu**[1,2,3]

[1]State Key Laboratory of Multimodal Artificial Intelligence Systems (MAIS), CASIA
[2]School of Artificial Intelligence, University of Chinese Academy of Sciences
[3]School of Information Science and Technology, ShanghaiTech University
[4]Center for Artificial Intelligence and Robotics, HKISI, CAS,
[5]People AI, Inc, [6]Stony Brook University
{zhangyiwei2023,gefudong2022,zhaoxiang.zhang}@ia.ac.cn,
{jin.gao,guanl,bli,wmhu}@nlpr.ia.ac.cn, hling@cs.stonybrook.edu

## Abstract

Bird's-eye-view (BEV) map layout estimation requires an accurate and full understanding of the semantics for the environmental elements around the ego car to make the results coherent and realistic. Due to the challenges posed by occlusion, unfavourable imaging conditions and low resolution, *generating* the BEV semantic maps corresponding to corrupted or invalid areas in the perspective view (PV) is appealing very recently. *The question is how to align the PV features with the generative models to facilitate the map estimation.* In this paper, we propose to utilize a generative model similar to the Vector Quantized-Variational AutoEncoder (VQ-VAE) to acquire prior knowledge for the high-level BEV semantics in the tokenized discrete space. Thanks to the obtained BEV tokens accompanied with a codebook embedding encapsulating the semantics for different BEV elements in the groundtruth maps, we are able to directly align the sparse backbone image features with the obtained BEV tokens from the discrete representation learning based on a specialized token decoder module, and finally generate high-quality BEV maps with the BEV codebook embedding serving as a bridge between PV and BEV. We evaluate the BEV map layout estimation performance of our model, termed VQ-Map, on both the nuScenes and Argoverse benchmarks, achieving 62.2/47.6 mean IoU for surround-view/monocular evaluation on nuScenes, as well as 73.4 IoU for monocular evaluation on Argoverse, which all set a new record for this map layout estimation task. The code and models are available on https://github.com/Z1zyw/VQ-Map.

## 1 Introduction

BEV layouts represent high-dimensional structured data that encompasses significant prior knowledge, particularly regarding road structures. While current methods for BEV map layout estimation mainly focus on constructing dense BEV features [2, 1, 3] for semantic segmentation as map prediction, they often overlook the incorporation of map prior knowledge. Additionally, occlusion and inherent challenges in depth estimation often lead to inaccuracies in dense features, especially in the areas that are corrupted or invalid in the PV. These factors contribute to incoherent and unrealistic BEV layout results, often with numerous artifacts (see Fig. 1). Yet, humans can rely solely on partial observations of a scene in the PV to imagine the entire coherent BEV layout elements. A natural

---

[*]Corresponding author

38th Conference on Neural Information Processing Systems (NeurIPS 2024).

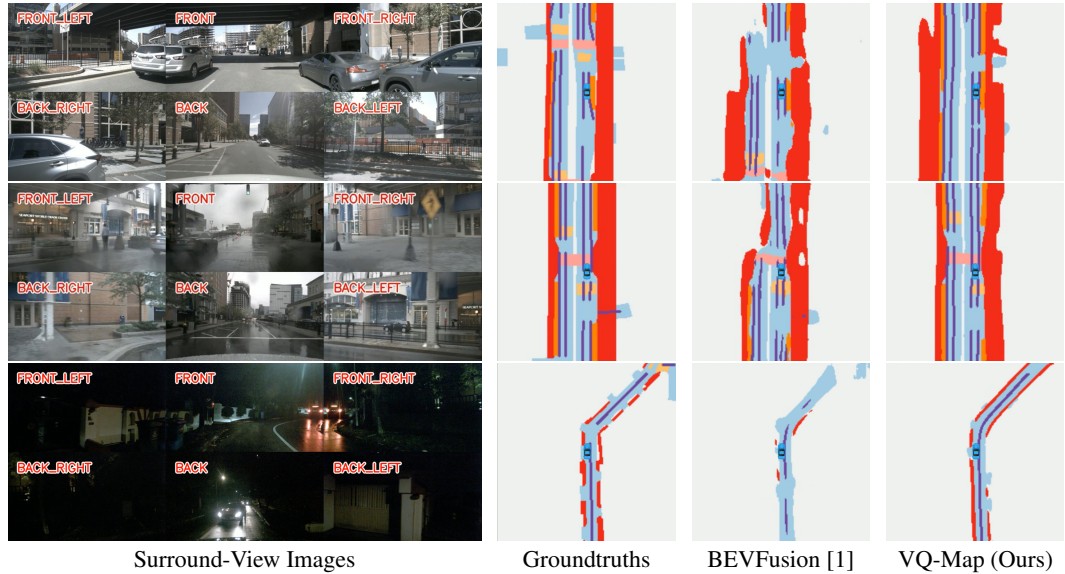

| Surround-View Images | Groundtruths | BEVFusion [1] | VQ-Map (Ours) |

Figure 1: We showcase the prediction results in various environmental conditions (day, rainy and night from top to bottom). Our VQ-Map produces more reasonable results, even for areas that are not directly visible, while significantly reducing artifacts. Color scheme is the same as in [1].

approach to imitating the human imagination process is to leverage generative models to learn the prior knowledge from the groundtruth BEV map layouts. However, ***the question is how to align the PV features with the generative models to facilitate BEV map estimation.***

To this end, we propose a novel pipeline called VQ-Map (see Fig. 2), which aligns the generative models well in the spirit of discrete tokens. In specific, VQ-Map utilizes a generative model similar to VQ-VAE [4] to encode the groundtruth BEV semantic maps into tokenized, discrete and sparse BEV representations, termed BEV tokens, accompanied with a discrete embedding space (*i.e.*, the codebook embedding). Each BEV token is the ***index*** of the nearest neighbor in the codebook embedding for an encoded BEV patch feature, representing the high-level semantics of a BEV patch. BEV tokens serve as a new classification label to directly supervise the PV feature learning via a specialized token decoder in our pipeline. The training of the generative model and the token decoder module is separated. By aligning with the sparse BEV tokens, our token decoder module is able to rely solely on sparse backbone features directly queried by token queries for BEV token prediction using an arbitrary transformer-like architecture [5–7]. Simultaneously, directly employing these sparse features for token prediction bypasses the challenges of building accurate dense BEV features in common practice. The predicted tokens can be integrated into BEV embeddings through the off-the-shelf codebook embedding for generating the final high-quality BEV semantic maps. This process is similar to the human brain's memory mechanism [8], where the targets (BEV map layouts) are encoded into highly abstract, sparse representations (BEV embeddings) through memory neurons (BEV tokens) that can be activated by specific visual signals (generated based on token queries).

We evaluate our proposed VQ-Map on both the surround-view and monocular map estimation tasks, and our method sets new records in both tasks, achieving 62.2/47.6 mean IoU for surround-view/monocular evaluation on nuScenes [9], as well as 73.4 IoU for monocular evaluation on Argoverse [10].

In summary, our contributions are as follows: **(1)** We propose a novel pipeline **VQ-Map** exploring a discrete codebook embedding to generate high-quality BEV semantic map layouts. The acquired prior knowledge subsequently helps to effectively align the sparse backbone image features with the generative models based on a specialized token decoder, leading to more accurate BEV map layout estimation with generation. **(2)** By formulating map estimation as the alignment of perception and generation, our achieved BEV codebook embedding serves as a bridge between PV and BEV, and can be used in the off-the-shelf manner. **(3)** Extensive experiments show that our VQ-Map establishes new state-of-the-art performance on camera-based BEV semantic segmentation. Meanwhile, we confirm that as a PV-BEV alignment method, token classification is more effective than value regression.

## 2   Related Work

**BEV Map Layout Estimation.** Most existing approaches treat BEV map layout estimation as a semantic segmentation task in the BEV frame, where map elements are rasterized into pixels with each allocated multiple class labels. As a pioneer of such technology, LSS [3] explicitly predicts discrete depth distributions on image features, and then 'lifts' these 2D features to obtain pseudo 3D features, which are finally flatten into BEV features through a pooling operation. Building upon it, BEVFusion [1] introduces LiDAR point clouds and implements multi-sensor fusion within a unified BEV space, effectively maintaining semantic and geometric information. Other approaches [11–15], such as VectorMapNet [12] and HIMap [15], tackle layout issues by incorporating vectorized prior maps. Besides, TaDe [16] utilizes a task decomposition strategy to improve monocular BEV semantic segmentation performance.

Recently, some methods utilize generative model-based technologies to enhance the performance of BEV map layout estimation. MapPrior [17] employs a generative map prior built on VQ-GAN [18] architecture to capture the detailed structure of traffic scenarios on the basis of conventional discriminative models, achieving a unified advantage in precision, realism and uncertainty awareness. Furthermore, DDP [19] and DiffBEV [20] focus on integrating the denoising diffusion process [21] into contemporary perception frameworks, exhibiting outstanding performance.

The above mentioned work MapPrior [17] and TaDe [16] both approach the BEV map segmentation task through two stages: a perceptual stage and a generative stage, which is relevant to our work. However, MapPrior aligns with the generative model by deriving complex BEV variables, which are constrained by the challenges of acquiring accurate dense BEV features. As for TaDe, training the generative model based on polar inverse-projected BEV groundtruth maps results in the loss of certain prior knowledge embedded in conventional BEV maps, making it prone to artifacts. In contrast, our method aligns the generative model with tokenized discrete representations, which are more meaningful and easier to predict, while also preserving BEV map prior knowledge.

**Tokenized Discrete Representation.** VQ-VAE [4] innovatively employs codebook mechanisms to establish an encoder-decoder architecture in a tokenized discrete latent space, capturing and representing richer and more complex data distributions. Following this approach, other generative models such as VQ-GAN [18], DALL-E [22] and VQ-Diffusion [23] also map inputs into discrete tokens corresponding to codebook entries to represent high-dimensional data. Meanwhile, some visual pre-training works [24, 25] use tokens to represent image patches and treats the prediction of masked tokens as a proxy task. Recently, UViM [26], Unified-IO [27] and AiT [28] encode various outputs as tokens and predicts them through an auto-regressive modeling [29], modeling a wide range of visual tasks. In this paper, we draw inspiration from the above work to predict BEV tokens for generating high-quality BEV map layouts.

## 3   Methods

We herein summarize our VQ-Map perception framework in Fig. 2 as follows. Firstly, we create the discrete representations which encapsulate the high-level BEV semantics for different BEV elements in the groundtruth maps to serve as the prior knowledge (*i.e.*, the codebook embedding) for map generation. Secondly, we conduct the PV-BEV alignment training with the specially designed token decoder module to predict the BEV tokens associated with the corresponding groundtruth maps. Finally, we directly combine the off-the-shelf codebook embedding accompanied by the map generation decoder with the PV-BEV alignment module to predict the BEV map layouts.

### 3.1   Discrete Representation Learning for BEV Generation

Similar to some visual pre-training methods [24, 25], we formulate the discrete representation learning as the task of BEV map reconstruction via a sequence of discrete tokens to acquire prior knowledge for the high-level BEV semantics. We obtain this tokenized discrete space by employing the VQ-VAE architecture [4], which comprises three modules: BEV Patch Embedding $\mathcal{E}$, Vector Quantization $\mathcal{Q}$ and BEV Map Generation Decoder $\mathcal{D}$. Roughly speaking, $\mathcal{E}$ transforms local BEV semantic patches into more abstract high-level semantics; $\mathcal{Q}$ then clusters the semantics derived from patch embedding to create the discrete representations; and finally, $\mathcal{D}$ is attached to utilize these discrete representations for reconstruction of the corresponding groundtruth maps.

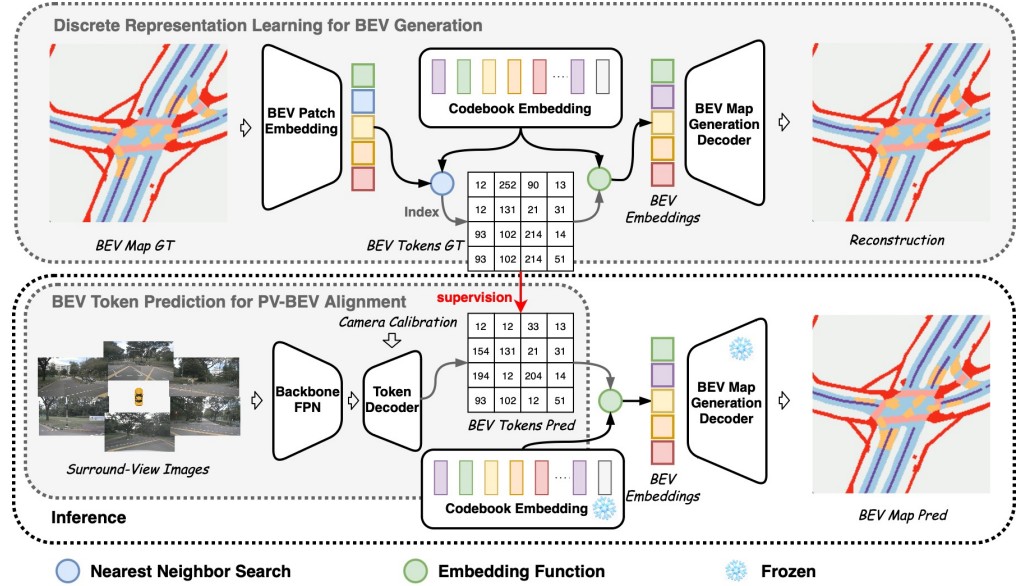

Figure 2: VQ-Map employs a generative model similar to the VQ-VAE framework to encode the BEV groundtruth maps into BEV tokens accompanied with a codebook embedding. After the generative model training, the BEV tokens serve as the classification labels to supervise the PV feature learning via a specialized token decoder module. During inference, VQ-Map utilizes the predicted BEV tokens to generate high-quality BEV map layouts based on the off-the-shelf codebook embedding and the BEV map generation decoder.

**BEV Patch Embedding $\mathcal{E}$.** BEV semantic maps significantly differ from the raw images with complex scenes. They inherently represent high-level semantics annotated by humans, which eliminates the need to aggregate extensive features using heavy encoders. Specifically, we initially patchify a groundtruth BEV map $\mathbf{M} \in \mathbb{B}^{C \times H \times W}$ into a sequence of non-overlapping BEV patches $\left\{\mathbf{M}^i \in \mathbb{B}^{C \times P \times P}\right\}_{i=1}^{N}$, where $\mathbb{B} = \{0, 1\}$, $P$ is the patch size, $C$ is the number of the groundtruth map layouts and $N = HW/P^2$ is the patch number. Our patch embedding $\mathcal{E}$ is simple, aiming to abstract high-level semantics $\mathbf{z}^i \in \mathbb{R}^D$ from individual patches $\mathbf{M}^i$, where $D$ is the embedded dimension. Fig. 3 shows some BEV patch images to visualize our discrete representation learning.

**Vector Quantization $\mathcal{Q}$.** We define a latent embedding space $\mathbf{V} \in \mathbb{R}^{K \times D}$ as our codebook embedding, where $K$ represents the maximum number of representations in the discrete latent space. We further denote it using the set $\{\mathbf{v}_1, \mathbf{v}_2, \ldots, \mathbf{v}_k, \ldots, \mathbf{v}_K\}$. Our vector quantization $\mathcal{Q}$ receives the continuous latent vector $\mathbf{z}_c$ from the patch embedding and outputs discrete latent $\mathbf{z}_q$, termed BEV embedding, through the nearest neighbor search in the codebook. This is calculated as

$$\mathbf{z}_q = \mathcal{Q}(\mathbf{z}_c) = \arg \min_{\ell_2(\mathbf{v}_k)} \|\ell_2(\mathbf{z}_c) - \ell_2(\mathbf{v}_k)\|_2 \tag{1}$$

where $\ell_2$ means L2 normalization employed for codebook lookup based on cosine similarity, as described in ImprovedVQGAN [30]. Each discrete latent can also be represented by its index in the codebook as the BEV token:

$$k_q = \arg \min_k \|\ell_2(\mathbf{z}_c) - \ell_2(\mathbf{v}_k)\|_2 \ . \tag{2}$$

**BEV Map Generation Decoder $\mathcal{D}$.** We feed the BEV embeddings $\{\mathbf{z}_q^i = \ell_2(\mathbf{v}_{k_q^i})\}_{i=1}^N$ to our map generation decoder $\mathcal{D}$ by firstly reshaping them into a grid format and then reconstructing the original groundtruth BEV map following

$$\mathbf{M}' = \mathcal{D}(\mathcal{Q}(\mathcal{E}(\mathbf{M}))) \ . \tag{3}$$

**Training Loss.** The overall training loss includes a VQ loss $\mathcal{L}_{vq}$ based on the codebook embedding due to the non-differentiable vector quantization operation besides the reconstruction loss $\mathcal{L}_{re}$. Unlike the common practice, we additionally incorporate a loss term reflecting the patch-level data augmentations (such as small-scale rotations, translations, and resizing) to aid clustering. The VQ

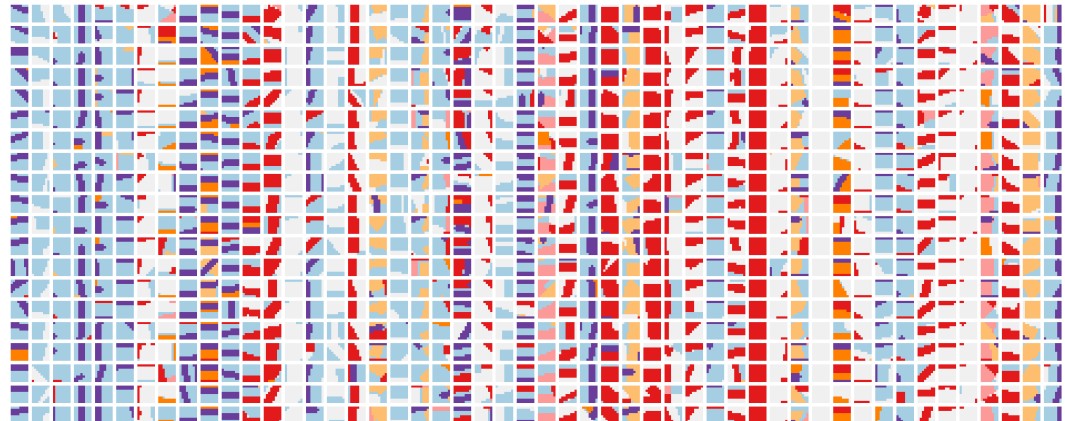

Figure 3: Visualization of the BEV codebook embedding by showing the BEV patch images corresponding to the specific BEV tokens. All BEV patch images in the same column correspond to the same token. The data is randomly sampled from the nuScenes validation dataset. Color scheme is the same as in [1].

loss is defined as:

$$\mathcal{L}_{vq} = \frac{1}{N} \sum_{i=1}^{N} \left( \left\| \mathbf{z}_q^i - \mathtt{sg}(\ell_2(\mathbf{z}_c^i)) \right\|_2^2 + \left\| \mathtt{sg}(\mathbf{z}_q^i) - \ell_2(\mathbf{z}_c^i) \right\|_2^2 + \sum_{j=1}^{N_{\text{aug}}} \left\| \mathtt{sg}(\mathbf{z}_q^i) - \ell_2(\tilde{\mathbf{z}}_c^{i,j}) \right\|_2^2 \right) \quad (4)$$

where $\mathtt{sg}$ denotes stop-gradient, $N_{\text{aug}} = 3$ denotes the number of patch augmentations, and $\tilde{\mathbf{z}}_c^i$ is the continuous latent vector derived from the augmented patch. We also use the exponential moving average (EMA) [4] to update the codebook embedding.

We utilize the class-specific weighted mean square error (MSE) for computing the reconstruction loss $\mathcal{L}_{re}$, where the weight of each class for each sample is inversely proportional to the number of groundtruth pixels in that class, $\mathbf{M}_c \in \mathbb{B}^{HW}$ and $\mathbf{M}_c' \in \mathbb{R}^{HW}$:

$$\mathcal{L}_{re} = \frac{1}{C} \sum_{c=1}^{C} \frac{\left\| \mathbf{M}_c - \mathbf{M}_c' \right\|_2^2}{1 + \left\| \mathbf{M}_c \right\|_1} \quad (5)$$

and the final loss $\mathcal{L}$ is defined as:

$$\mathcal{L} = \mathcal{L}_{re} + \mathcal{L}_{vq} . \quad (6)$$

### 3.2 Token Prediction with Sparse Features for PV-BEV Alignment

Once the codebook embedding $\mathbf{V}$ is obtained from the vector quantization $\mathcal{Q}$ and BEV patching embedding $\mathcal{E}$, the dense prediction task of BEV semantic segmentation for the map layouts, represented by $\mathbf{M} \in \mathbb{B}^{C \times H \times W}$, can be transformed into a sparse token classification task represented by a grid $\overline{\mathbf{M}} \in \mathbb{A}^{(H/P) \times (W/P)}$ consisting of BEV tokens (see Eq. (2)) as illustrated in Fig. 2, where $\mathbb{A} = \{1, 2, 3, \dots, K\}$.

Since each token represents a BEV patch instead of a BEV pixel as mentioned in Sec. 1, the BEV tokens are significantly sparser compared to the dense BEV features in common practice and the final BEV semantic maps. Each BEV semantic token can be recognized based on the semantics at the corresponding position in the image features. So we propose a transformer-like token decoder module based on deformable attention [6] to query image semantic features at the individual position to predict the corresponding BEV token. This approach is inspired by the transformer-based objection detection methods in DETR [7], Deformable-DETR [6] and BEVFormer [2].

**Token Decoder Module.** Our token decoder consists of multiple patch-level cross-attention layers, a convolutional layer and a head as shown in Fig. 4. It takes multi-scale image backbone features from the feature pyramid network (FPN) [31] and the camera calibration as input, and outputs the predicted sparse BEV tokens for PV-BEV alignment. We use $N$ learnable embeddings as our token queries, each associated with a specific 3D position determined via the LiDAR coordinate system. Specifically, the token reference points (or anchors in the following) are initially set in the LiDAR coordinates and subsequently projected to image space using the camera and LiDAR calibration.

For each patch-level cross-attention layer, it follows the classical decoder layer architecture [5] with self-attention, cross-attention and feed forward network (FFN). Initially, token queries engage in self-attention interactions, restricted to neighboring queries. Then, multiple anchors are set for each query, and the deformable cross-attention [6] samples features only from the images of surround view that the anchors can be projected onto. When the anchors of a query involve multiple images, the average of the sampled features from these images is taken [2]. Given that each query aims to capture patch-level semantics, the anchors are positioned at varying heights, widths, and depths before being projected onto the images. Finally, refined token queries are obtained through a FFN.

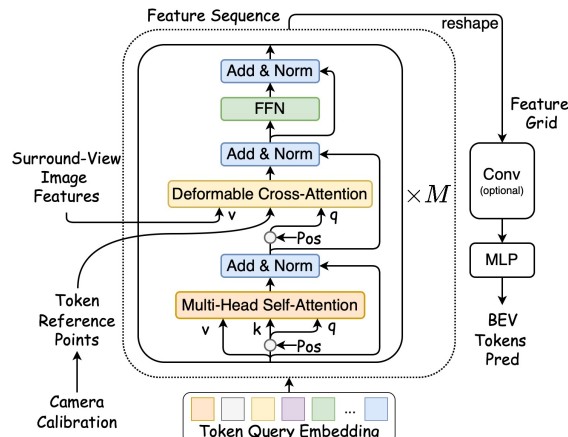

Figure 4: **Architecture of Our Token Decoder**. *Pos* refers to the positional embedding, and $M$ indicates the layer number.

After multiple patch-level cross-attention layer, a feature sequence is obtained and reshaped to a feature grid for predicting BEV tokens. We use a convolutional layer to integrate the neighboring sparse features in the feature grid. After the convolutional layer, there is a multi-layer perceptron (MLP) serving as the classification head to predict tokens. We utilize focal loss [32] to optimize this classification task.

## 4 Experiments

We evaluate the performance of our VQ-Map on both the surround-view and monocular BEV map layout estimation tasks and use the Intersection-over-Union (IoU) metric for evaluation.

We perform the surround-view experiments on nuScenes [9] involving 1000 on-road scenes from four locations in Boston and Singapore. Each scene has a duration of approximately 20 seconds, resulting in a total of 40k samples/key-frames. Each sample is captured by six monocular cameras, covering a 360° panoramic view around the ego vehicle. For BEV map estimation, nuScenes provides manual annotations encompassing 11 layout classes. According to the prior studies [3, 1], we train and validate VQ-Map on 700 scenes with 28130 samples and 150 scenes with 6019 samples respectively. In addition, we transform the original map layouts into the ego-centric frame through rasterization. We evaluate the IoU scores for 6 map layout classes, *i.e.*, drivable area, pedestrian crossing, walkway, stop line, car-parking area, and lane divider. Additionally, we calculate the mean IoU (mIoU) averaged over all of them.

We utilize both the nuScenes [9] and Argoverse [10] datasets for the monocular task following the common practice. In particular, Argoverse employs seven surround-view cameras for data collection. It comprises 65 training sequences and 24 validation sequences captured in Miami and Pittsburgh. Like the nuScenes dataset, Argoverse provides detailed and comprehensive semantic map layout annotations. We evalutate the IoU scores for drivable area, pedestrian crossing, walkway and car-parking area on nuScenes while only the drivable area on Argoverse.

### 4.1 Implementation Details for Surround-View Task

In this section, we primarily outline the experimental settings for the surround-view task on nuScenes, while leaving the settings for the monocular task in Appendix A due to the space limitation.

**Data Preparation.** Following BEVFusion [1], we generate the BEV semantic segmentation map within a square area surrounding the ego car, which spans from -50 meters to 50 meters along both the x and y axes. We set the segmentation resolution at 0.5 meters per pixel, which culminates in a final image output of 200 by 200 pixels in size. Considering the possibility of overlapping classes within the map, our model is designed to carry out binary segmentation across all classes. The input images are resized to 256×704.

Table 1: State-of-the-art comparison for the surround-view BEV map layout estimation on the nuScenes **validation** set. MapPrior [17] uses a fixed IoU threshold of 0.5, while other methods apply the threshold that maximizes IoU according to their original settings. In our method, we adopt a constant IoU threshold of 0.5 to ensure a fairer comparison across all existing approaches. We only evaluate different approaches in the camera-only setting.

| Methods | IoU ↑ (%) | | | | | | |
| --- | --- | --- | --- | --- | --- | --- | --- |
| | Drivable | Ped. Cross. | Walkway | Stopline | Carpark | Divider | Mean |
| OFT [36] | 74.0 | 35.3 | 45.9 | 27.5 | 35.9 | 33.9 | 42.1 |
| LSS [3] | 75.4 | 38.8 | 46.3 | 30.3 | 39.1 | 36.5 | 44.4 |
| CVT [37] | 74.3 | 36.8 | 39.9 | 25.8 | 35.0 | 29.4 | 40.2 |
| M$^2$BEV [38] | 77.2 | - | - | - | - | 40.5 | - |
| BEVFusion [1] | 81.7 | 54.8 | 58.4 | 47.4 | 50.7 | 46.4 | 56.6 |
| MapPrior [17] | 81.7 | 54.6 | 58.3 | 46.7 | 53.3 | 45.1 | 56.7 |
| X-Align [34] | 82.4 | 55.6 | 59.3 | 49.6 | 53.8 | 47.4 | 58.0 |
| MetaBEV [35] | 83.3 | 56.7 | 61.4 | 50.8 | 55.5 | 48.0 | 59.3 |
| DDP [19] | 83.6 | 58.3 | 61.6 | 52.4 | 51.4 | 49.2 | 59.4 |
| VQ-Map | **83.8** | **60.9** | **64.2** | **57.7** | **55.7** | **50.8** | **62.2** |

**Discrete Representation Learning Details.** We use a BEV patch size $P = 8$ and the codebook embedding of $K = 256$ with a dimensionality of $D = 128$. Each sample of the semantic groundtruth map can be divided into $N = 25 \times 25 = 625$ patches. For BEV patch embedding, we employ three $3 \times 3$ convolutional modules without padding, with channels of $64, 128$, and $128$ respectively. A $2 \times 2$ max pooling layer is inserted before the final convolutional module. Following VQGAN [18], we construct the BEV map generation decoder using 5 layers with each including 3 residual blocks. Additionally, we append a sigmoid function at the end of the decoder. Regarding to the codebook embedding optimization, we set the EMA decay to 0.99 and the epsilon value to 1e-5.

**Token Decoder Details.** We adopt SwinT-tiny [33] as our backbone to maintain consistency with the prior work [1, 17, 19, 34, 35]. The token decoder comprises 8 patch-level cross-attention layers and a convolutional module with a kernel size of 5. Each layer has a dimension of 512, with 8 heads and 16 reference anchors sampled from 4 heights, 2 depths and 2 widths. Each anchor predicts 2 offsets for each scale of the features map. Self-attention is performed for each token interaction within a $5 \times 5$ region. Additionally, we set the output channel number of the FPN to 512 to align with the token decoder. During the inference stage, we also use the soft token technology [28] to represent the BEV embeddings.

**Training.** For the discrete representation learning, we employ an initial learning rate of 1e-4 and a batch size of 16 per GPU, and conduct training on 2 NVIDIA A100 (40G) GPUs, totaling 35 GPU hours. Subsequently, for training PV-BEV alignment, we adjust the initial learning rate to 5e-5 and the batch size to 8 per GPU for training on 4 NVIDIA A100 (40G) GPUs, totaling 96 GPU hours. The training is conducted over 20 epochs, incorporating warm-up and cosine learning rate decay strategies. Additionally, we apply a weight decay factor of 0.01.

## 4.2 State-of-the-Art Comparison

We conduct the state-of-the-art comparison for the surround-view BEV map layout estimation task by comparing to several recent competitive methods as shown in Tab. 1. Among them, BEVFusion [1] is widely regarded as a standard framework that integrates three key components, *i.e.*, a backbone module, a view transformation and a task decoder. X-Align [34] uses perspective supervision to integrate perspective predictions based on dense BEV feature for enhanced performance. Map-Prior [17] employs a generative model to enhance the perceptual outcomes derived from BEVFusion. Meanwhile, MetaBEV [35] introduces an innovative cross-attention module subsequent to the dense BEV feature extraction, which is instrumental in addressing sensor failures. Additionally, DDP [19] leverages a denoising diffusion process [21] within its decoder to achieve better precision.

Tab. 1 shows that our VQ-Map sets a new mIoU performance record of 62.2 for the surround-view map estimation task on nuScenes when comparing to the above methods. In particular, VQ-Map achieves a 5.5 mIoU gain in comparison to BEVFusion, with a notable improvement of over 10 in the *Stopline* layout class. VQ-Map also consistently outperforms all other methods by significant

Table 2: State-of-the-art comparison for the monocular BEV map layout estimation on the nuScenes and Argoverse **validation** sets using the IoU (%) metric. Our VQ-Map uses the IoU threshold of 0.5 while other methods choose the best threshold following their original settings. During the evaluation process, grid cells that cannot be reached by LiDAR are ignored [39].

| Methods | nuScenes [9] | | | | | Argoverse [10] |
| | Drivable | Crossing | Walkway | Carpark | Mean | Drivable |
|---|---|---|---|---|---|---|
| IPM [39] | 40.1 | - | 14.0 | - | - | 43.7 |
| Depth Unpr. [39] | 27.1 | - | 14.1 | - | - | 33.0 |
| VED [40] | 54.7 | 12.0 | 20.7 | 13.5 | 25.2 | 62.9 |
| VPN [41] | 58.0 | 27.3 | 29.4 | 12.9 | 31.9 | 64.9 |
| PON [39] | 60.4 | 28.0 | 31.0 | 18.4 | 34.5 | 65.4 |
| DiffBEV [20] | 65.4 | 41.3 | 41.1 | 28.4 | 44.1 | - |
| GitNet [42] | 65.1 | 41.6 | 42.1 | 31.9 | 45.2 | 67.1 |
| TaDe [16] | 65.9 | 40.9 | 42.3 | 30.7 | 45.0 | 68.3 |
| VQ-Map | **70.0** | **43.9** | **43.8** | **32.7** | **47.6** | **73.4** |

margins, with a minimum mIoU gain of 2.8 without relying on any other supervisions like PV semantic segmentation or depth estimation.

For the monocular BEV map layout estimation task, we replace the backbone of our VQ-Map with ResNet50 [43] to align with the previous work [39, 42] (Please refer to Appendix A for more implementation details). As shown in Tab. 2, our VQ-Map also consistently outperforms all other competitive methods that are publicly available on the nuScenes and Argoverse datasets with significant margins. In comparison to the second best approach on nuScenes GitNet [42], we achieve a 2.4 mIoU performance gain, while a 5.1 IoU gain on Argoverse in comparison to the second best TaDe [16]. We provide visualization results in Fig. A3. *It is noteworthy that our experiments on the Argoverse [10] dataset directly exploit the off-the-shelf codebook embedding and BEV map generation decoder trained on nuScenes, indicating good transferability of our discrete representation learning.*

### 4.3 Ablations and Analysis

Table 3: Ablation experiments on some key parameters of the token decoder. We perform ablations on the token decoder layer number $M$ using layer dimension of 512, and ablations on different layer dimension by setting $M$ to 8.

(a) Ablation for $M$ of token decoder.

| $M$ | 2 | 4 | 6 | 8 |
|---|---|---|---|---|
| Drivable | 81.1 | 82.7 | 83.6 | **83.8** |
| Ped. Cross. | 55.9 | 58.2 | 60.1 | **60.9** |
| Walkway | 59.2 | 61.7 | 63.5 | **64.2** |
| Stop Line | 50.9 | 55.1 | 56.8 | **57.7** |
| Carpark | 49.9 | 52.2 | **56.2** | 55.7 |
| Divider | 47.3 | 49.0 | 50.3 | **50.8** |
| Mean | 57.4 | 59.8 | 61.8 | **62.2** |

(b) Ablation for the layer dimension of token decoder.

| *Layer Dimension* | 256 | 512 | 768 |
|---|---|---|---|
| Drivable | 83.0 | **83.8** | 82.9 |
| Ped. Cross. | 58.4 | **60.9** | 57.9 |
| Walkway | 62.4 | **64.2** | 61.6 |
| Stop Line | 54.5 | **57.7** | 54.1 |
| Carpark | 53.6 | **55.7** | 52.5 |
| Divider | 48.5 | **50.8** | 48.6 |
| Mean | 60.1 | **62.2** | 59.6 |

**Token Decoder Parameters.** The number of patch-level cross-attention layers $M$ in the token decoder has a substantial impact on the final performance, as evidenced in Tab. 3a. We note that as $M$ increases, the mIoU value also increases. However, the mIoU improvement becomes marginal when $M$ increases from 6 to 8, and we thus use 8 layers by default. Additionally, we investigate the effects of setting different dimensions for the token decoder layers in Tab. 3b, which shows that the optimal dimension is 512. Note that the following ablation studies all employ 6 layers and 512 dimension.

**Dense *vs.* Sparse BEV Features.** In the token decoder of our approach, the features before the classification MLP head can also be regarded as a kind of sparse BEV features with shape $25 \times 25 \times 512$, obtained based on deformable attention. In this ablation, we compare the utilization of dense BEV features *vs.* sparse BEV features for predicting either the sparse tokens based on the off-the-shelf codebook embedding or directly the dense semantic maps, where the dense BEV feature is obtained through LSS [3] following BEVFusion with shape $128 \times 128 \times 80$.

Table 4: Ablation for various design choices. Comparing (a) (b) (c) (f) entails assessing different architecture designs, while (d) (e) (f) involves comparing different supervisions during the PV-BEV alignment training. The supervision signals shown in columns (d), (e) and (f) are the explored three kinds of intermediate results in Sec. 3.1. The differences are: (d) uses latent variables that have not been discretized by the codebook, (e) uses the latent variables that have been discretized by the codebook, and (f) uses the codebook indices. (g) sets $N_{aug} = 0$ in Eq. (4).

| | | (a) | (b) | (c) | (d) | (e) | (f) | (g) |
|---|---|---|---|---|---|---|---|---|
| Arch. | Sparse Feature | | | ✓ | ✓ | ✓ | ✓ | ✓ |
| | Codebook Embedding | | ✓ | ✓ | ✓ | ✓ | ✓ | ✓ |
| Supervision | | $\mathbf{M}$ | $\{k_q^i\}_{i=1}^N$ | $\mathbf{M}$ | $\{\mathbf{z}_c^i\}_{i=1}^N$ | $\{\mathbf{z}_q^i\}_{i=1}^N$ | $\{k_q^i\}_{i=1}^N$ | $\{k_q^i\}_{i=1}^N$ |
| Drivable | | 81.5 | 80.4 | **83.9** | 82.5 | 82.5 | 83.6 | 83.5 |
| Ped. Cross. | | 54.2 | 52.9 | 60.0 | 59.7 | 59.1 | **60.1** | 59.9 |
| Walkway | | 58.1 | 58.2 | 63.5 | 62.2 | 62.1 | **63.5** | 63.4 |
| Stop Line | | 46.1 | 47.2 | 53.2 | 55.1 | 54.9 | **56.8** | 56.8 |
| Carpark | | 53.2 | 52.7 | 51.0 | 53.4 | 53.1 | **56.2** | 55.1 |
| Divider | | 45.3 | 46.6 | 46.9 | 48.9 | 49.0 | 50.3 | **50.7** |
| Mean | | 56.4 | 56.3 | 59.8 | 60.3 | 60.1 | **61.8** | 61.6 |
| Improvements | | | -0.1 | 3.4 | 3.9 | 3.7 | **5.4** | 5.2 |

Table 5: Computational overhead analysis. Training time is measured in GPU hours using NVIDIA A100 (40G). Our method, even in its tiny version, surpasses the previous SOTA DDP (3 steps). Additionally, the computational cost (MACs) of our tiny version is significantly lower than previous methods. For the standard version of the model, it achieves substantial performance improvements while maintaining a relatively low computational cost.

| Method | mIoU↑(%) | Params↓(M) | MACs↓(G) | Training Time↓(h) |
|---|---|---|---|---|
| BEVFusion | 56.6 | 50.1 | 155.5 | **100** |
| MapPrior | 56.7 | 719.1 | 396.0 | >200 |
| DDP(3 steps) | 59.4 | 53.6 | 614.1 | 160 |
| VQ-Map(tiny) | 59.6 | **44.2** | **86.8** | 30+74=104 |
| VQ-Map(light) | 60.1 | 81.9 | 137.3 | 35+80=115 |
| VQ-Map | **62.2** | 108.3 | 231.6 | 35+96=131 |

To predict dense maps using sparse BEV features in (c), we replace the classification MLP head in the token decoder with BEVFusion's map decoder and the groundtruth BEV map $\mathbf{M}$ is used for supervision. (a) can be seen as a variant of BEVFusion. Conversely, for predicting sparse tokens using dense BEV features in (b), we introduce a deformable decoder [6] after the dense BEV features that initialized with weights from BEVFusion. VQ-Map (f) predicts sparse tokens using sparse BEV features. The results in Tab. 4 show that sparse BEV features are more effective for predicting sparse tokens than dense BEV features, and our achieved sparse features work better even in the traditional end-to-end framework for the map estimation task.

**Different PV-BEV Alignment.** This part of the experiment explores which kind of intermediate results of our discrete representation learning based on VQ-VAE (see Sec. 3.1) is best for the second stage training of PV-BEV alignment. Instead of using the sparse token classification task for PV-BEV alignment supervision, we also test the case of utilizing the continuous latent variables directly derived from the patch embedding as a regression task for supervision in (d). In (e), we test an alternative strategy for supervision which involves predicting the latent BEV features and obtaining the BEV embeddings through nearest neighbor search based on the codebook embedding. In both (d) and (e), MSE is employed to train the token decoder. As shown in Tab. 4, the performance supervised by BEV tokens using focal loss surpasses that supervised by latent variables using MSE. These results indicate that BEV tokens serve as a superior abstract representation for guiding PV-BEV alignment.

**Computational Overhead Analysis.** We provide the computational overhead for our VQ-Map on the surround-view task as shown in Tab. 5, including the number of parameters, computational cost (MACs), and model training time. Decreasing the layer dimension of the token decoder from 512 to 256 resulting in a light version of our VQ-Map. Since the learned discrete representations also require a significant number of parameters, we further use a smaller architecture with $K = 128$, $D = 64$ in $\mathcal{Q}$ and $\mathcal{D}$ to achieve a tiny version of our VQ-Map. It shows that even the tiny version of our approach can still achieve comparable results to the recent SOTA methods in Tab. 1. Our approach not only demonstrates strong performance, but also saves much computational cost in comparison to the recent

SOTA methods MapPrior and DDP, in both the training and testing phases. In addition, the two-stage training of our approach introduces some additional training overhead in comparison to BEVFusion.

## 5 Conclusion

In this paper, we present a novel pipeline VQ-Map by aligning with the generative models using discrete BEV tokens, which efficiently enhances the performance of BEV map layout estimation. The core components of our method are the codebook embedding constructed via vector quantization as prior knowledge, serving as a bridge between PV and BEV, and the specially designed BEV token decoder to facilitate the PV-BEV alignment, both of which enable the generation of high-quality BEV semantic map layouts. We hope that our work will inspire further research on vector quantization, providing assistance not only for map estimation and its downstream tasks, but also for a wide range of other applications.

**Limitations and future work.** A significant limitation of our approach is our inability to handle semantics that are position-sensitive and have small areas. It is challenging for patch-level semantics to effectively represent position information for small-scale semantics using the architecture similar to VQ-VAE. Tokenization in our approach is more robust against noise and geometric changes. However, it may lead to a loss of some detailed spatial information. When the dangerous holes and some random obstacles appear in the realistic AD environment, an anomaly detection module may be needed to handle this situation. Our approach is a specialized method only for BEV map generation at present, but we believe the token-based multi-task modeling for autonomous driving is very promising. Additionally, tokenized intermediate results are well-suited for combining with large language models.

**Broader Impacts.** Accurate BEV layout estimation helps drivers and autonomous driving systems better understand the surrounding environment, thereby reducing the occurrence of traffic accidents and improving traffic safety. However, it may be misused to capture detailed information about the environment, including sensitive data such as building layouts, infrastructure, and so on.

**Acknowledgements.** This work was supported in part by the Beijing Natural Science Foundation (Grant No. L223003, JQ22014), the Natural Science Foundation of China (Grant No. U22B2056, 62422317, 62192782, 62036011, U2033210, 62102417, 62222206, U2033210, 62172413), the Project of Beijing Science and technology Committee (Project No. Z231100005923046). Jin Gao and Bing Li were also supported in part by the Youth Innovation Promotion Association, CAS. Haibin Ling was not supported by any fund for this work.

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

# Appendix

## A   Implementation Details for the Monocular Experiments

**Data Preparation.** Following PON [39], we define a square region in front of the given monocular camera, spanning $[0, 50] \times [-25, 25]$ meters. Setting the resolution to 0.25 meters per pixel results in a final map size of $200 \times 200$ pixels during the evaluation. Additionally, the images are resized to $600 \times 800$ for nuScenes and $600 \times 960$ for Argoverse. We perform the monocular experiments using all available surrounding cameras with training/validation splits outlined in PON [39].

**Details of the Monocular Layout Estimation Model Training.** The groundtruth map is resized to $224 \times 224$ during our model training, and the size of the BEV patch is set to $16 \times 16$, resulting in a total of $14 \times 14 = 196$ patches. We use ResNet50 as the backbone to align with the previous work [39] and employ a token decoder with 6 patch-level cross-attention layers and 256 layer dimension.

For the discrete representation learning, we set the initial learning rate to 1e-4 with a batch size of 32 per GPU, and conduct training on 2 NVIDIA A100 (40G) GPUs for 50 epochs, totaling 28 GPU hours. Subsequently, for training PV-BEV alignment, we adjust the initial learning rate to 5e-4 and the batch size to 8 per GPU for training on 4 NVIDIA A100 (40G) GPUs for 40 epochs, totaling 42/44 GPU hours for nuScenes/Argoverse. We also resize our output map size from $224 \times 224$ back to $200 \times 200$ using a maxpooling operation to aligns with the previous work's results during the evaluation. Since our VQ-Map only predicts BEV tokens within the view frustum range, zero padding is used for latent representations in other areas. When calculating the focal loss, the patches that LiDAR cannot reach will be ignored.

## B   More Visualization Results

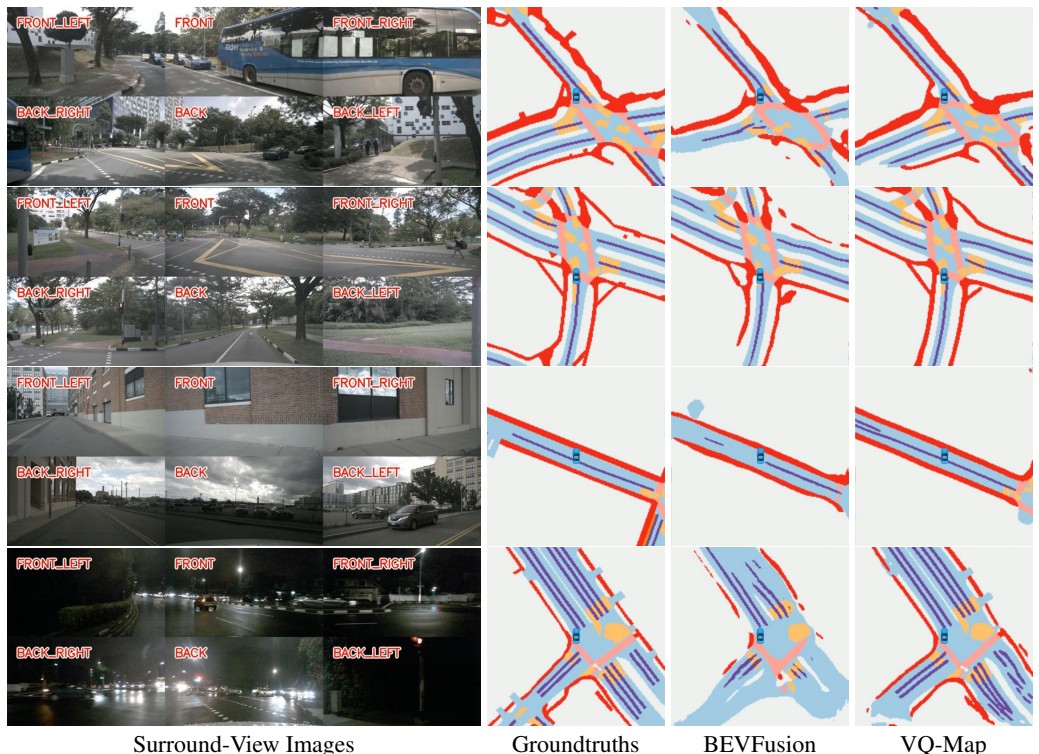

| Surround-View Images | Groundtruths | BEVFusion | VQ-Map |

Figure A1: More visualization results for sorround-view BEV map layout estimation on nuScenes.

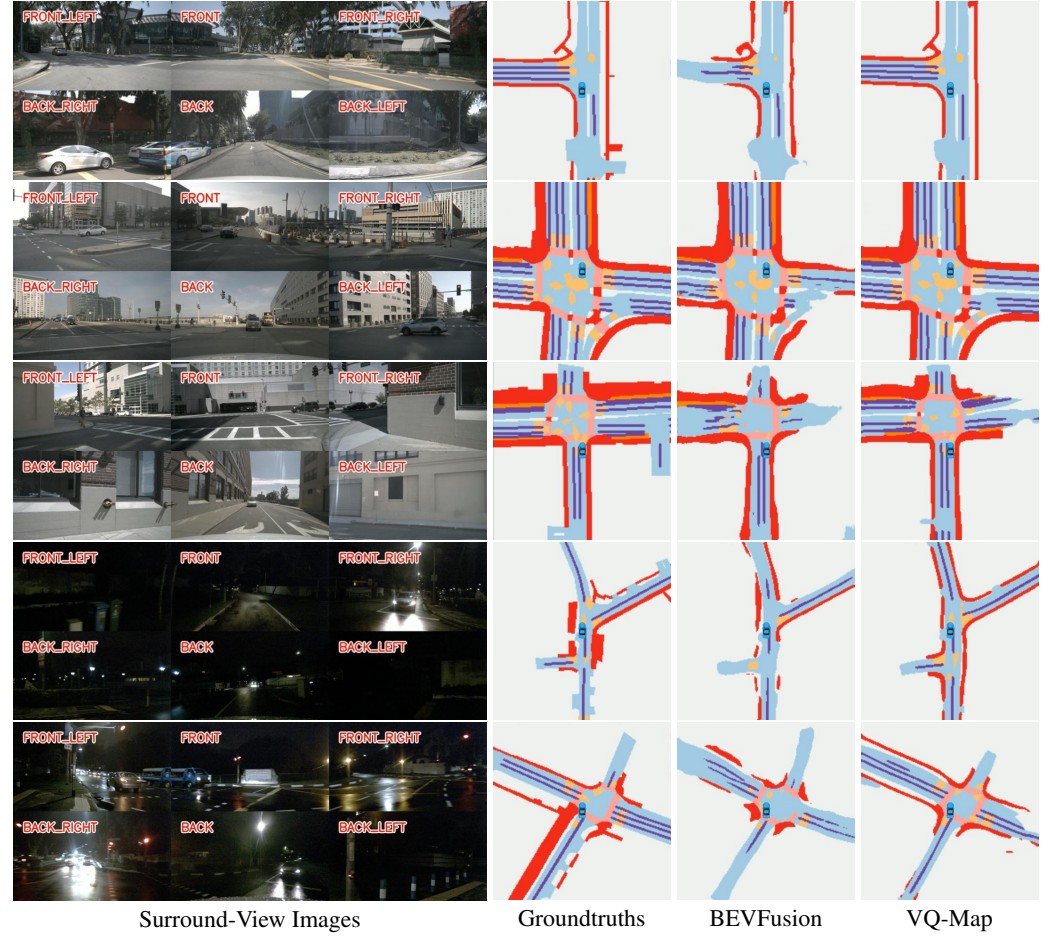

Surround-View Images     Groundtruths     BEVFusion     VQ-Map

Figure A2: More visualization results for sorround-view BEV map layout estimation on nuScenes.

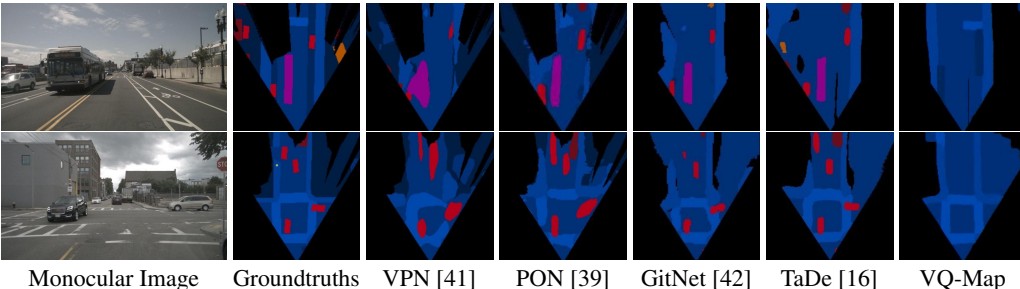

Monocular Image   Groundtruths   VPN [41]   PON [39]   GitNet [42]   TaDe [16]   VQ-Map

Figure A3: Visualization results for monocular BEV map layout estimation on nuScenes. Unlike other methods, our VQ-Map only focuses on map layout classes. The color scheme is the same as in PON [39].

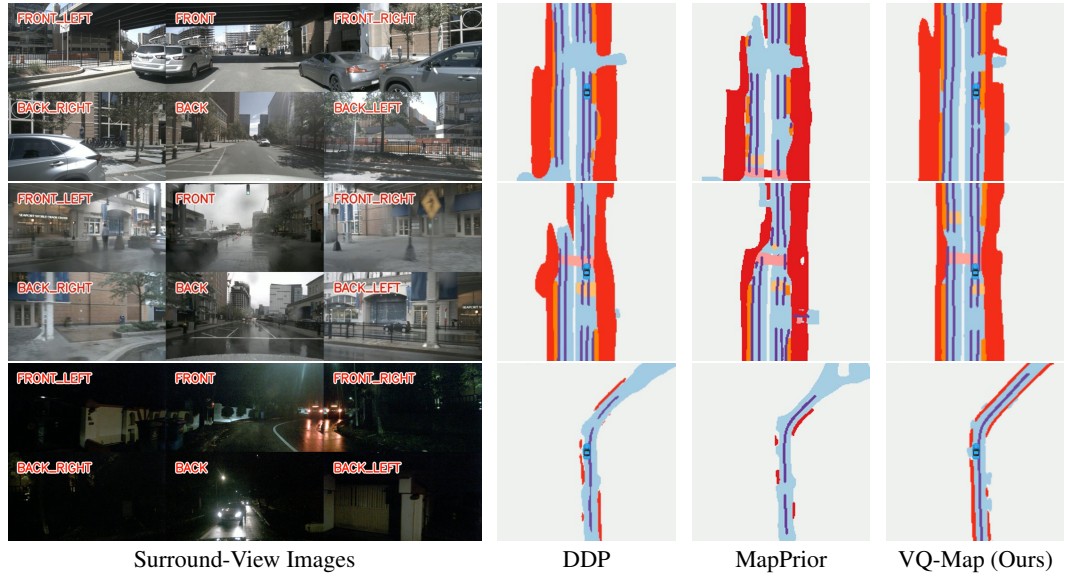

| Surround-View Images | DDP | MapPrior | VQ-Map (Ours) |

Figure A4: More visualization results for sorround-view BEV map layout estimation on nuScenes. We showcase the comparison of our method with DDP and MapPrior in various environmental conditions (day, rainy and night from top to bottom), corresponding to Fig. 1

## C  More Experiments

Table A1: Ablation experiments on some key parameters of the codebook embedding. We perform ablations on the codebook size $K$ using dimension $D$ of 128, and ablations on different dimension $D$ by setting $K$ to 256.

(a) Ablation for $K$ of codebook embedding.

| $K$ | 128 | 256 | 512 |
| --- | --- | --- | --- |
| Drivable | 83.5 | **83.6** | 83.5 |
| Ped. Cross. | **61.3** | 60.1 | 60.0 |
| Walkway | 63.4 | **63.5** | 63.3 |
| Stop Line | **57.9** | 56.8 | 57.2 |
| Carpark | 53.0 | **56.2** | 55.5 |
| Divider | 50.2 | 50.3 | **50.5** |
| Mean | 61.5 | **61.8** | 61.7 |

(b) Ablation for $D$ of codebook embedding.

| $D$ | 64 | 128 | 256 |
| --- | --- | --- | --- |
| Drivable | **83.7** | 83.6 | 83.6 |
| Ped. Cross. | 59.9 | **60.1** | 59.9 |
| Walkway | 63.4 | **63.5** | 63.2 |
| Stop Line | 56.6 | **56.8** | 56.3 |
| Carpark | 55.6 | **56.2** | 54.5 |
| Divider | 50.2 | **50.3** | 49.9 |
| Mean | 61.6 | **61.8** | 61.2 |

**Cdoebook Embedding Parameters.** We conduct the experiments with different codebook size $K \times D$, and the results in Tab. A1a and Tab. A1b show that our proposed method is insensitive to the size of the codebook in a wide range.

Table A2: More Metrics. Chamfer distance is used to evaluate the boundary quality of the drivable area, corresponding to Drivable IoU. Normalized IoU and Mean IoU (test set) further illustrate the superiority of the method. MMD is a metric capturing the realism in scene structures.

| Metric | BEVFusion | DDP | VQ-Map |
| --- | --- | --- | --- |
| Cham. Dis. ↓ (Pixels) | 2.67 | 2.45 | **2.21** |
| Drivable IoU ↑ (%) | 81.7 | 83.6 | **83.8** |
| Normalized IoU ↑ (%) | 70.6 | 72.9 | **74.2** |
| Mean IoU (test set) ↑ (%) | 63.9 | 67.2 | **70.2** |
| MMD (val. set) ↓ | 39.6 | **23.7** | 24.4 |
| MMD (test set) ↓ | 19.5 | 12.9 | **10.0** |

**Boundary Quality Evaluation for Drivable Area.** While IoU is widely used to assess the performance of BEV map predictions, it primarily reflects the overall overlap between predicted and groundtruth areas. However, it does not fully capture the accuracy of the boundary regions, particularly in tasks like drivable area layout estimation, where boundary precision is crucial. We introduced an additional metric to specifically evaluate drivable area boundary quality.

We applied the Sobel operator to detect boundaries in the predicted and groundtruth BEV maps, followed by calculating the chamfer distance between them. The chamfer distance, measured in pixels, provides a more precise evaluation of boundary alignment (lower values indicate better boundary matching). In Tab. A2 we compare our approach to BEVFusion and the previous SOTA method DDP for surround-view drivable area layout estimation. While our model achieves a modest 0.2% improvement in IoU over DDP, the Chamfer distance decreases by an average of 0.24 pixels. This improvement indicates that our method estimates the drivable area boundaries more accurately than the previous methods, further supporting the effectiveness of our approach for BEV map estimation.

**Normalized IoU.** Considering the ped. cross., walkway, stopline, carpark and divider is much fewer than the drivable area, we compute average performance based on normalization of the raw pixels, *i.e.*Normalized IoU. The results in Tab. A2 shows our method still maintains an advantage.

**Evaluation in test set.** In earlier BEV semantic segmentation works [3, 39], pixel-wise segmentation was applied to map both background static objects and foreground dynamic objects. However, because the nuScenes test set does not provide ground truth for dynamic objects, these works could not use it to evaluate their methods. To maintain consistency with prior work, we adopt the same dataset split as LSS [3] for the surround-view task, using the validation set for evaluation. For the monocular task, we follow the setup of the pioneering work PON [39], whose code provides a calibration set for tuning hyper-parameters and uses a re-divided validation set for evaluation. We also tested the off-the-shelf trained models from BEVFusion, DDP and our approach on the nuScenes test set to further show the superiority of our approach (although none of the previous works conduct the comparison on the nuScenes test set). Our approach in Tab. A2 consistently outperforms DDP and BEVFusion by large margins like in Tab. 1.

**About the realism (MMD) results.** MMD is a metric of distance between the generated layout predictions and the ground truth distribution, which is to capture the realism in scene structures [17]. It is noteworthy that this realism metric and the common used precision metric are not closely coupled. It is possible to achieve higher IoU while generating non-realistic map layouts, or vice versa. MMD of MapPrior in nuScenes validation set is 28.4 [17]. We provide the MMD comparison between BEVFusion, DDP and our approach under the camera-only setting in Tab. A2, which is conducted on both the nuScenes validation set and test set. It shows that the generative prior models can enjoy the better structure preservation ability, and our approach VQ-Map is the best at pushing the limit of both the precision and realism simultaneously.

**About providing the uncertainty awareness (ECE) results.** In MapPrior, its generative stage after the predictive stage during inference introduces a discrete latent code $\mathbf{z}'$ to guide the generative synthesis process through a transformer-based controlled synthesis in the latent space, where multiple diverse $\mathbf{z}^{(k)}$ are obtained using nucleus sampling and then decoded into multiple output samples (see Figs. 2 and 5 in [17]). So it is necessary to evaluate the uncertainty awareness (ECE) score for the generated multiple layout estimation samples. As for the discriminative BEV perception baselines that are trained end-to-end with cross-entropy loss, the predictions produced by the softmax function are assumed to be the pseudo probabilities to facilitate the computation of ECE score.

However, during inference of our VQ-Map, the token decoder outputs the classification probabilities for each BEV token, and our approach enables us to use these probabilities as weights to perform a weighted sum of the elements in the codebook embeddings and use it as input to the decoder to achieve one final layout estimation output. This process is similar to the 1-step version of MapPrior, in which there is no need to evaluate the ECE score as shown in Tab. 1 of [17].

**Different Attention in Token Decoder.** We also tested alternative design by replacing the deformable attention in (f) with standard attention. Its mean IoU is 56.5, lower than that of (f), which indicates that deformable attention is more suitable for this task. We designed the token decoder module based on deformable attention thanks to its ability to model the prior positional relationship between PV and BEV for feature alignment. Deformable attention can naturally use reference points to leverage these priors, whereas standard attention cannot.

