# OpenReview forum: "VQ-Map: Bird's-Eye-View Map Layout Estimation in Tokenized Discrete Space via Vector Quantization"
_NeurIPS.cc/2024/Conference — NeurIPS 2024 poster_

### Official Review · Reviewer_1JdE · 2024-06-18

**Soundness:** 3
**Presentation:** 3
**Contribution:** 2
**Rating:** 6
**Confidence:** 4

**Summary:**

To address challenges facing generating BEV maps posed by occlusion, unfavorable imaging conditions, and low resolution, this paper introduces a generative model to facilitate BEV estimation. A codebook embedding is used to encoder prior knowledge for the high-level BEV semantics in the tokenized discrete space. Compared with other methods leveraging dense features to supervise BEV maps, this paper takes this discrete semantic feature as a signal to supervise BEV tokens learned from PV views. In a word, the article is logical and clear in writing. Moreover,  it has excellent results and sufficient ablation experiments.

**Strengths:**

This paper is well-written and easy to understand. Moreover, the experimental data is rich. Both surround-view and monocular map estimation tasks are assessed. The proposed model has superior accuracy results compared to other methods. The visualization result of the BEV codebook is quite interesting.

**Weaknesses:**

In the experiment, the parameters of the ablation method and the table were not explained clearly.

In Table 4, the difference between (a), (b) and (c) is not clearly described.

The specific meaning of 'Supervision' in (d) - (f) is not described, which should be added.

**Questions:**

1. The motivation of this paper is unclear. Why do sparse features work better than dense features, or do they work better together?
2. In 3.2, the description is not clear. Does this part need supervision? If so, how do you implement the supervision method?
3. The results of Table 3 show that 8-layer and 512-dimensional  are a better choice. Doubtly, compared to other methods (6-layer and 256-dimensional),  the proposed method has better results due to the increased number of layers and dimensions. The comparison is somewhat unfair. Would you consider discussing this?
4. Would you consider presenting more computational complexity results like FLOPs/MACs, the number of parameters, and running time when comparing your proposed method against existing methods in the experiment tables, e.g., Table 1 and Table 2?
5. Would you consider presenting a set of visual comparisons to directly compare your proposed method against MapPrior? This can better show the advantage of your proposed solution.

**Limitations:**

This paper has well analyzed the limitations and broad implications of the study.

---

> ### Author Rebuttal · Authors · 2024-08-07
>
> Thank you for your review and feedback!
>
> **About clearly explaining the parameters of the ablation method and the table**: We apologize for not providing more details for the ablation methods in Tab. 4. The parameters that vary across the different ablation methods mainly include the layer and dimension number settings that depend on whether our deformable attention-based architecture is used or not, and the feature spatial size and dimension that depend on whether the dense or sparse BEV features are used. The supervision signals for training PV-BEV alignment also vary across the different ablation methods. We have included more explainations for each of them when asking the subsequent questions. We will also integrate the clarification for them in the revision.
>
> **About clearly describing the difference between (a), (b) and (c)**: The specific structures of (a), (b) and (c) are briefly described in Lines 276-281. We are sorry for not describing them clearly. Columns (a) and (b) both use the dense BEV features generated from the BEVFusion method while not using the attention-like architecture like in our token decoder module. As (a) directly predicts the BEV maps, it can be seen as a variant of BEVFusion. (b) uses dense features to predict the sparse BEV tokens and finally generate the BEV maps. (c) uses the sparse features generated from our proposed deformable attention-based architecture to directly predict the BEV maps like the traditional end-to-end methods. We will include more details for them in the revision.
>
> **About clearly describing the specific meaning of 'Supervision' in (d), (e) and (f)**: This part of the experiment explores which kind of intermediate results of our discrete representation learning based on VQ-VAE (see Sec. 3.1) is best for the second stage training of PV-BEV alignment. The supervision signals shown in columns (d), (e) and (f) are the explored three kinds of intermediate results. The differences are: (d) uses latent variables that *have not* been discretized by the codebook, (e) uses the latent variables that *have* been discretized by the codebook, and (f) uses the codebook indices. It shows that token classification (f) is the most effective method for PV-BEV alignment. We will add this description into the caption of Tab. 4.
>
> **About the motivation of using sparse features in our token decoder**: In the second stage training for PV-BEV alignment in our framework, the BEV tokens generated from the BEV groundtruth maps are used as the supervision signals as illustrated in Fig. 2. Whether using the dense or sparse BEV features for token decoder can be experimentally determined. We first found that using dense features with 128 $\times$ 128 $\times$ 80 dimensions to predict the sparse BEV tokens in (b) even performs worse than the traditional end-to-end method in (a). Considering that using sparse features to align with our tokenization idea may be more robust against noise and geometric changes (as acknowledged by Reviewer `xhZa`), we thus tested using sparse features with 25 $\times$ 25 $\times$ 512 dimensions to predict the sparse BEV tokens based on our proposed deformable attention-based architecture, which significantly improves the overall performance (comparing (f) to (b)). We also surprisingly found that our achieved sparse features work better even in the traditional end-to-end framework (comparing (c) to (a)) for the BEV map estimation task.
>
> Due to the failure of using dense features in our experiments, it may be hard to get better performance when using sparse and dense features together (the performance degrades to 61.3 mIoU). However, we experimentally find that training the sparse features to predict tokens can improve the training of dense features which are used to directly predict the maps. Its performance increases from 56.4 to 57.7 mIoU.
>
>
> **About the description in 3.2**: This part is designed for predicting BEV tokens, and the supervision signals during the second stage training for PV-BEV alignment in our framework are the BEV tokens generated from the BEV groundtruth maps as illustrated in Fig. 2. We cast it as a classification task and use the focal loss (Lines 188-189) to train the backbone FPN and token decoder. This part does not involve any additional supervisions. We will clarify this point in the revision.
>
> **About the different layer and dimension number settings in the ablation experiment of Tab. 4**: In the ablation experiment of Tab. 4, columns (a) and (b) both use the dense BEV features generated from the BEVFusion method (not using the attention-like architecture like ours), so comparing our best result 62.2 mIoU using 8-layer and 512-dimensional settings in column (f) to them may be OK. However, columns (c) (d) (e) and (g) all use 6-layer and 512-dimensional settings, it may be not appropriate to compare (f) with them. We apologize for this confusion. We will replace the results for column (f) with 61.8 mIoU results using 6-layer and 512-dimensional settings (see Tab. 3) in the revision.
>
> **About more computational complexity results**: Following your suggestion, we have added a computational complexity comparison using the number of parameters, MACs, and the training time. Please refer to Tab. R1 in the attached PDF. It clearly shows that our approach not only demonstrates strong performance (also acknowledged by Reviewer `1hbx`), but also saves much computational cost in comparison to the recent SOTA methods MapPrior and DDP, in both the training and testing phases. In addition, the two-stage training of our approach introduces some additional training overhead in comparison to BEVFusion. We will include them in the revision.
>
>
> **About more visualization comparisons against MapPrior**: Thank you for your suggestion. We show more visualization comparisons against MapPrior and DDP in Fig. R1 of the attached PDF, which can better show the advantage of our proposed solution. We will include them in the revision.

---

> > ### Comment · Reviewer_1JdE · 2024-08-08
> > **Comment**
> >
> > The rebuttal helps to address many of the concerns. The added computation complexity analysis and qualitative visualization comparison results should be integrated into the final version.
> >
> > MapPrior also provides realism (MMD) and uncertainty awareness (ECE) results. Would it be possible to provide such results for analysis as well? This could better verify the superiority of your approach over the baseline method.
> >
> > Sincerely,

---

> > > ### Author Response · Authors · 2024-08-09
> > > **Thank Reviewer 1JdE for the reply**
> > >
> > > Thank you for your careful review and thoughtful reply, and we are happy to continue the discussion. The added computation complexity analysis and qualitative visualization comparison results will be integrated into the final version.
> > >
> > > **About providing the realism (MMD) results**: MMD is a metric of distance between the generated layout predictions and the ground truth distribution, which is to capture the realism in scene structures (e.g., the lane and sidewalks that have gaps and are not straight may result in topological changes and drastically impact downstream modules). It is noteworthy that this realism metric and the common used precision metric are not closely coupled. It is possible to achieve higher IoU while generating non-realistic map layouts, or vice versa. We provide the MMD comparison between BEVFusion, MapPrior, DDP and our approach under the camera-only setting in the table below, which is conducted on both the nuScenes *validation* set and *test* set. It shows that the generative prior models can enjoy the better structure preservation ability, and our approach VQ-Map is the best at pushing the limit of both the precision and realism simultaneously. We will include this MMD-based analysis in the revision.
> > >
> > > |            | BEVFusion | MapPrior | DDP | VQ-Map |
> > > | --------- | --------- | --------- | --------- | --------- |
> > > | MMD$\downarrow$ on *validation* set | 39.6 | 28.4 | 23.7 | 24.4 |
> > > | MMD$\downarrow$ on *test* set | 19.5 | - | 12.9 | 10.0 |
> > >
> > >
> > > **About providing the uncertainty awareness (ECE) results**: In MapPrior[17], its generative stage after the predictive stage during inference introduces a discrete latent code $\textbf{z}’$ to guide the generative synthesis process through a transformer-based controlled synthesis in the latent space, where multiple diverse $\textbf{z}^{(k)}$ are obtained using nucleus sampling and then decoded into multiple output samples (see Figs. 2 and 5 in [17]). So it is necessary to evaluate the uncertainty awareness (ECE) score for the generated multiple layout estimation samples. As for the discriminative BEV perception baselines that are trained end-to-end with cross-entropy loss, the predictions produced by the softmax function are assumed to be the pseudo probabilities to facilitate the computation of ECE score.
> > >
> > > However, during inference of our VQ-Map, the token decoder outputs the classification probabilities for each BEV token, and our approach enables us to use these probabilities as weights to perform a weighted sum of the elements in the codebook embeddings and use it as input to the decoder to achieve **one** final layout estimation output. This process is similar to the 1-step version of MapPrior, in which there is no need to evaluate the ECE score as shown in Tab. 1 of [17]. We will clarify this point in the revision.

---

> > > > ### Comment · Reviewer_1JdE · 2024-08-11
> > > > **Comment**
> > > >
> > > > The reviewer would like to thank the authors for supplementing these analyses, which could be added to the revised version.
> > > >
> > > > Sincerely,

---

### Official Review · Reviewer_xhZa · 2024-07-12

**Soundness:** 2
**Presentation:** 3
**Contribution:** 2
**Rating:** 3
**Confidence:** 5

**Summary:**

The authors propose to use a generative model similar to the Vector Quantized-Variational AutoEncoder (VQ-VAE) to obtain prior knowledge for high-level Bird’s Eye View (BEV) semantics in a tokenized discrete space. By leveraging BEV tokens and a codebook embedding that encapsulates the semantics for different BEV elements, the proposed method aligns image features with the BEV tokens obtained from discrete representation learning in a token decoder. The experiments are conducted on two benchmarks, nuScenes and Argoverse.

**Strengths:**

+ The paper is well-written and very easy to follow. The figures are also very intuitive, significantly facilitating other researchers to follow and understand the gist.

+ The visualization of the proposed method helps readers understand how the proposed method works.

+ In fact, I quite like the tokenization idea since tokenization may be more robust against noise and geometric changes. However, tokenization may lose some information or detailed spatial information. Considering the output of this task, the output layout is much coarser than the input though. So tokenization may work well.

**Weaknesses:**

At a glance, the proposed method achieves very good performance. In the experimental part, the proposed method outperforms the competing method. However, there are two main issues in the experiments:

**The competing methods are not state-of-the-art**
If we look at the reference [17] published in ICCV 2017, that is the state-of-the-art method (The same task and the same dataset). Moreover, the code of [17] is also publicly available. Comparison with reference [19] which is a generative model is not compelling. Based on the reported results of [17], the results of the proposed method are much worse than that of [17].

**The dataset split of nuScene is not right**
In nuScene, there are 700 scenes for training, 150 scenes for validation and 150 scenes for testing. In this paper, (L197-198) there is no testing dataset. The authors mentioned to validate the performance of the proposed method. It is not clear whether the authors use the validation set or testing set. Moreover, how do the authors select the model weights if a validation set is not provided?

**Questions:**

My biggest concern is the dataset split for this work. I expect the authors can explain this clearly.

---

> ### Author Rebuttal · Authors · 2024-08-06
>
> Thank you for your review and feedback! We are happy that you quite like our tokenization idea for the BEV map layout estimation task. We provide responses to the specific points below:
>
> **About the state-of-the-art comparison with MapPrior[17] and DDP[19]**: Both of the two references of MapPrior[17] and DDP[19] are published at ICCV'2023, and evaluate their approaches in both the camera-only setting and multi-modality setting (with LiDAR information combined). In both settings, the mIoU of DDP always surpasses MapPrior, i.e., 59.4 vs 56.7 in the camera-only setting and 70.6 vs 63.1 in the multi-modality setting. In Tab. 1 of our state-of-the-art comparison, we report the results for MapPrior[17] and DDP[19] only using their published results under the camera-only setting, because our state-of-the-art comparison is completely based on the camera-only setting. *We are very sorry for not clearly introducing the camera-only setting used in our main paper.* We will remendy this misleading issue in the revision. In Tab. 1, it clearly shows that our VQ-Map can achieve the best performance in comparison to other entries under the camera-only setting. We will also delve into how to incorporate data from other modalities (like the LiDAR) into our framework in the future work.
>
> **About the dataset split of nuScenes in the comparison**: In the much earlier BEV semantic segmentation works, e.g., LSS[3] published at ECCV'2020 and PON[39] at CVPR'2020, the pixel-wise segmentation is conducted for the layout of both the static objects in the background and dynamic objects in the foreground. However, the nuScenes *test* set does not provide the groundtruth for the dynamic objects in the foreground. So these works directly use the nuScenes *validation* set to validate the performance of their proposed approaches. To align with these pioneering works and make the fair comparison, the subsequent works shown in Tabs. 1 and 2 all keep using this dataset split setting by default, where the models are evaluated on the nuScenes *validation* set. More specifically, for our evaluation in the surround-view experiments in Tab. 1, only the *training* set is used to train our model and the number of training epochs and total iterations is aligned with the previous work BEVFusion. The model weights from the last epoch are used for final evaluation on the nuScenes *validation* set. For our evaluation in the monocular experiments in Tab. 2, we follow the practice of the pioneering work PON, which re-divides the (*training*, *validation*) split into (*training*, *calibration*, *validation*) split, where the *calibration* set is used to adjust the hyper-parameters, and the *validation* set is used for evaluation. We will clearly explain this dataset split setting in the revised version.
>
> Furthermore, to ease the concern with regard to the nuScenes *test* set, where the groundtruth for the layout of the static objects in the background is available, we directly tested the off-the-shelf trained models from BEVFusion, DDP and our approach on the nuScenes *test* set to further show the superiority of our approach (although none of the previous works conduct the comparison on the nuScenes *test* set). As shown in the table below, our approach consistently outperforms DDP and BEVFusion by large margins like in Tab. 1.
>
> |                    | BEVFusion | DDP  | VQ-Map   |
> | ------------------ | --------- | ---- | -------- |
> | mIoU$\uparrow$ (%) | 63.9      | 67.2 | **70.2** |

---

### Official Review · Reviewer_rd8m · 2024-07-17

**Soundness:** 3
**Presentation:** 3
**Contribution:** 2
**Rating:** 6
**Confidence:** 4

**Summary:**

This paper proposes to use a generative model to encode BEV semantic maps into tokenized sparse BEV representations with codebooks.
Specifically, it consists a two stage training scheme. First, train a BEV generation VQ-VAE, then use the BEV Tokens as a ground truth to train the second stage network where it maps the surround view images into BEV map. The second stage reuse the first stage BEV generator. Experiments show the performance surpass the state-of-the-art by 0.2% - 3% in IoU, on average improving 2.8% IoU on nuScenes.

**Strengths:**

+ This paper proposes a complicated pipeline to generate BEV maps, using a pre-trained VQ-VAE as a intermediate representation then trains a two stage network to firstly encode features from perspective view images into the BEV map.
+ The idea of using a pre-trained network as codebook seems quite interesting in the BEV map generation domain

**Weaknesses:**

Q1. On average, is IoU a good metric for measuring the performance of BEV map? Considering the road can be a continuous structure, probably one of the most important aspect to measure the drivable area is its boundary, is this faithfully reflected in the IoU metric?

Q2. I think the mean IoU increase is summing over the different settings, but not in terms of their raw pixel right? for example, in Table 1, there are 6 factors, f1 to f6, where each f is computed based on their raw pixels. Then the mean is computed by sum{f1, ..., f6} / 6. However, considering the Walkway, stopline, carpark and divider is much fewer than the drivable area, can you compute the average performance on based on normalization of  their raw pixels as well?

Q3 Complexity of the cost

Can the authors list their methods complexity? Since it consists of two stage training and might introduce many overhead. Though it is not a big deal, but it is also interesting to the community.

**Questions:**

as above

**Limitations:**

This paper cannot handle small regions but might be very important to realistic AD environment, where holes and some random obstacles can be dangerous to the vehicle.
Compared to other method that can achieve multiple task at the same time, this is a specialized method only for BEV map generation.

---

> ### Author Rebuttal · Authors · 2024-08-06
>
> Thank you for your review and feedback! We are happy that you find our idea of using a pre-trained codebook interesting in the BEV map generation domain. We provide responses to the specific points below:
>
> **About the IoU metric for measuring the performance of BEV map**: The common practice of using the IoU metric to reflect the overlap between the models' predictions and the groundtruths can indicate the overall accuracy of the predictions to a certain extent. However, we agree with you that the drivable area boundary is also one of the most important aspects to measure the accuracy of the predictions, and the IoU metric may not faithfully reflect the accuracy of the drivable area boundary. Inspired by your valuable comments, we thus propose to use the Sobel operator to detect the boundaries of the models' predictions and evaluate the boundary quality using the Chamfer distance metric in pixels (the smaller, the better). We compare our method with BEVFusion and the previous SOTA aproach DDP for the surround-view drivable area layout estimation in the table below. Compared to DDP, although our IoU score for the drivable area only achieves 0.2% performance gain, the Chamfer distance for our approach decreases by an average of 0.24 pixels, indicating that our approach can estimate the drivable area layout with more accurate boundary. We will include this new analysis in the revision.
>
> | Metric                          | BEVFusion | DDP  | VQ-Map   |
> | ------------------------------- | --------- | ---- | -------- |
> | Cham. Dis.$\downarrow$ (Pixels) | 2.67      | 2.45 | **2.21** |
> | IoU$\uparrow$ (%)               | 81.7      | 83.6 | **83.8** |
>
> **About computing the average performance based on normalization of the raw pixels**: For each background class in the map layout estimation, an IoU score is evaluated based on the raw pixels and reported in percentage. The common practice of computing mIoU is to take the average of the IoU scores across all the evaluated background classes as you have pointed out. Following your suggestion, we compute the average performance (Normalized IoU) over all the evaluated background classes based on normalization of the raw pixels as well and show the comparison with BEVFusion, MapPrior and DDP in the table below. We will include this new analysis in the revision.
>
> |                              | BEVFusion | MapPrior | DDP  | VQ-Map   |
> | ---------------------------- | --------- | -------- | ---- | -------- |
> | Normalized IoU$\uparrow$ (%) | 70.6      | 70.6     | 72.9 | **74.2** |
>
> **About the complexity of the cost**: Following your suggestion, we have added a computational complexity comparison using the number of parameters, MACs, and the training time. Please refer to Tab. R1 in the attached PDF for “global” response. It clearly shows that our approach not only demonstrates strong performance (acknowledged by Reviewer `1hbx`), but also saves much computational cost in comparison to the recent SOTA methods MapPrior and DDP, in both the training and testing phases. In addition, the two-stage training of our approach introduces some additional training overhead in comparison to BEVFusion. We will include this computational complexity analysis in the revision.
>
> **About handling small regions**: As acknowledged by Reviewer `xhZa`,  tokenization in our approach is more robust against noise and geometric changes, which however may lose some information or detailed spatial information. So our approach cannot handle small regions well. This may be the inherent limiation of our approach due to the No Free Lunch rule. When the dangerous holes and some random obstacles appear in the realistic AD environment, an anomaly detection module may be needed to handle this situation. We will detail this discussion in the limitation section.
>
> **About achieving multiple tasks at the same time**: Although our approach is a specialized method only for BEV map generation at present, we believe the token-based multi-task modeling for autonomous driving is very promising. Additionally, tokenized intermediate results are well-suited for combining with large language models. We will detail this discussion in the conclusion and leave the token-based multi-task modeling in the future work.

---

> > ### Comment · Reviewer_rd8m · 2024-08-13
> > **Thanks**
> >
> > I have no further question and raised my score.

---

### Official Review · Reviewer_1hbx · 2024-07-21

**Soundness:** 3
**Presentation:** 3
**Contribution:** 2
**Rating:** 5
**Confidence:** 3

**Summary:**

The paper proposes a novel approach, VQ-Map, for BEV map layout estimation, addressing the challenges of occlusion and low-resolution images in perspective views. By leveraging a VQ-VAE-like generative model, the authors introduce BEV tokens to bridge the gap between sparse image features and dense BEV representations. The method demonstrates strong performance on both nuScenes and Argoverse benchmarks, setting new state-of-the-art results.

**Strengths:**

The paper clearly identifies the challenges in BEV map layout estimation and provides a well-defined solution.


The proposed method is based on the use of generative models like VQ-VAE for BEV map layout estimation and aims to address the limitations of existing methods.


The proposed method achieves good results on nuScenes and Argoverse benchmarks.

**Weaknesses:**

The proposed idea of using generative model is based on the previous methods like DDP or DiffBEV.


The proposed method like toke-based decoder is not very novel for the feature alignment. For example, the deformable attention is used and explored by many other methods for feature alignment.


The paper focuses on two specific datasets. The comparison with more recent diffusion methods is not included.

**Questions:**

How about the proposed method using other attention instead of the deformable attention?



What is the effect of the codebook size?



In Table 1, what is the reason to make two different settings for the experiment results in nuScenes validation set?

**Limitations:**

The proposed methods are based on the previous methods like DDP or DiffBEV. The technical contributions are not very novel, like the use of the deformable attention.

---

> ### Author Rebuttal · Authors · 2024-08-06
>
> Thank you for your review and feedback! We are happy that you recognized that our paper clearly identifies the challenges in BEV map layout estimation and provides a well-defined solution with a novel approach VQ-Map proposed. We provide responses to the specific points below:
>
> **About the idea of using generative model**: DDP and DiffBEV propose an end-to-end framework to generate a more comprehensive BEV representation and denoise noisy samples by *applying the diffusion model* to BEV perception. Although we share the similar idea of using generative model, our approach achieves better performance with much lower MACs (see Tab. R1 in the attached PDF for the comparison with DDP) as the diffusion process may introduce a significant computational overhead. We will clarify this point in the revision.
>
> **About the novelty with regard to our token-based decoder and using other attention**: In the main paper, we have pointed out (Lines 41-43) that our token decoder module can be an arbitrary transformer-like architecture to be compatible with our novel pipeline VQ-Map. Following the common practice, we design this token decoder module based on deformable attention thanks to its ability to model the prior positional relationship between PV (Perspective View) and BEV (Bird's Eye View) for feature alighment. Deformable attention can naturally use reference points to leverage these priors. Following your suggestion, we also consider other two kinds of attention in our token-based decoder for ablation comparison on the nuScenes validation set, i.e., replacing the deformable attention with (1) the standard attention and (2) the cross view attention proposed in CVT[37], respectively. We experimentally find that the deformable attention performs better than the above considered alternatives as shown in the table below. We will include this new ablation comparison in the revision.
>
> | **Attention** | Deformable | Standard (1) | Cross View (2) |
> | ------------- | ---------- | ------------ | -------------- |
> | **mIoU**      | 61.8       | 56.5         | 49.0           |
>
>
> **About the comparison with more recent diffusion methods**: In Sec. 4.2 of our main paper, we have conducted the comparison with the most recently published diffusion-based BEV map layout estimation works, i.e., DDP[19] and DiffBEV[20] published in ICCV'2023 and AAAI'2024 respectively. In the surround-view comparison in Tab. 1, we report the results of DDP[19] using its camera-only modality results published in the original paper. In the monocular comparison in Tab. 2, we report the published results of DiffBEV[20] based on its original paper. In Tab. R1 of the attached PDF, we have also presented a computational complexity comparison with DDP to better show the superiority of our approach. Note that we are not able to conduct the computational complexity comparison with DiffBEV due to the fact that its publicly available code is not complete. We will detail the analysis in comparison to DDP and DiffBEV in the revision.
>
> **About the effect of codebook size**: We have conducted the experiments with different codebook size $K\times D$, and the results (shown in the table below) show that our proposed method is insensitive to the size of the codebook in a wide range. We will include this analysis in the revision.
>
> | $K~(D=128)$ | 128  | 256  | 512  |
> | ------------ | ---- | ---- | ---- |
> | **mIoU**     | 61.5 | 61.8 | 61.7 |
>
> | $D~(K=256)$ | 64  | 128  | 256  |
> | ------------ | ---- | ---- | ---- |
> | **mIoU**     | 61.6 | 61.8 | 61.2 |
>
> **About the different IoU threshold settings for different methods compared in Tab. 1**：Most of previous methods publish the results following the evaluation protocol in BEVFusion[1], where the threshold that maximizes IoU is used, while MapPrior[17] uses the same IoU threshold of 0.5 across all the background classes for the BEV map layout estimation. We believe using a constant IoU threshold for all the background classes in our approach can deliver a more fair comparison with the existing approaches without suppressing the performance of the methods that use the threshold maximizing IoU. Note that our approach using the threshold maximizing IoU achieves 62.3 mIoU in comparison to 62.2 in Tab. 1. We will clarify this point in the revision.

---

> > ### Comment · Reviewer_1hbx · 2024-08-11
> > **Questions**
> >
> > Thanks for the response. The rebuttal has addressed many of my concerns.
> >
> > An interesting point is that using deformable attention to replace standard attention can lead to such a large improvemence. Which of the other methods (in Tab. 4 ablation study) has or has not included deformable attention? How about if include? Apart from that, is it possible to list the respective improvements from different proposed components?

---

> > > ### Author Response · Authors · 2024-08-13
> > > **Thank Reviewer 1hbx for the reply**
> > >
> > > Thank you for your careful review and thoughtful reply, and we are sorry for not clearly describing the specific structures and meaning of the ablation methods in Tab. 4, though they are briefly described in Lines 276-281.
> > >
> > > **About the deformable attention in the ablation study of Tab. 4**: The supervision signals during the second stage training for PV-BEV alignment in our novel pipeline are the BEV tokens generated from the BEV groundtruth maps as illustrated in Fig. 2. It shows that token classification (f) using the codebook indices is the most effective method for PV-BEV alignment in comparison to columns (d) and (e), which are corresponding to using latent variables that *have not* and *have* been discretized by the codebook respectively. Column (g) aims to show the effectiveness of $N_{aug}$ in Eq. (4). They all use the sparse features generated from our proposed deformable attention-based architecture to predict the supervision signals.
> > >
> > > Whether using the dense or sparse BEV features for token decoder in our pipeline can be experimentally determined. Columns (a) and (b) both use the dense BEV features generated from the BEVFusion method while not using the attention-like architecture like in our token decoder module. We first found that using dense features with 128$\times$128$\times$80 dimensions to predict the sparse BEV tokens in (b) even performs worse than the traditional end-to-end method in (a). Note that (a) directly predicts the BEV maps and it can be seen as a variant of BEVFusion. Considering that using sparse features to align with our tokenization idea may be more robust against noise and geometric changes (as acknowledged by Reviewer `xhZa`), we thus tested using sparse features with 25$\times$25$\times$512 dimensions to predict the sparse BEV tokens based on our proposed deformable attention-based architecture, which significantly improves the overall performance (comparing (f) to (b)). We also surprisingly found that our achieved sparse features work better even in the traditional end-to-end framework (comparing (c) to (a)) for the BEV map estimation task.
> > >
> > > Based on the above analyses, we can see that only (a) and (b) do not have the deformable attention included. If (a) uses our achieved sparse features based on the proposed deformable attention-based architecture, it becomes (c) with mIoU improved from 56.4 to 59.8. Due to the failure of using dense features (see column (b) with 56.3 mIoU) or the standard attention-based sparse features (56.5 mIoU), our novel pipeline really enjoys the benefit of deformable attention to achieve the new performance record of 62.2 mIoU, which significantly surpasses the SOTA method DDP (59.4 mIoU). Note that deformable attention has always been a common practice for PV-BEV alignment in the discriminative BEV perception approaches that are trained end-to-end with cross-entropy loss. We will include more details for them in the revision.
> > >
> > > **About the respective improvements from different proposed components**: Our generative pipeline mainly includes two components, i.e., the deformable attention-based token decoder using sparse features and the codebook embedding for BEV generation, where some parameter settings include the layer and dimension number settings for the deformable attention-based architecture, and the supervision signals for training PV-BEV alignment after discrete representation learning. We list the respective improvements for them below:
> > > - The deformable attention-based sparse features employing 6 layers and 512 dimensions improves over the baseline (a) to achieve (c) with 3.4 mIoU performance gain. We also tested the deformable attention-based dense features with 128$\times$128$\times$80 dimensions, which ahcieves 1.6 mIoU performance gain over (a).
> > > - The codebook embedding using the latent variables as the supervision signals further improves over (c) to achieve (d) or (e) with 0.3-0.5 mIoU performance gains, while using the codebook indices achieves 61.8 mIoU with 2.0 mIoU performance gain over (c).
> > > - Further changing the layer and dimension number settings to 8 layers and 512 dimensions finally achieves (f) with an additional mIou performance gain of 0.4.
> > >
> > > We will add these respective improvements into Tab. 4 for more clear presentation.

---

### Author Rebuttal · Authors · 2024-08-07

Thank you for the valuable reviews pointing out that our *novel*  (`1hbx`) approach and the idea of a pre-trained codebook are particularly *interesting* (`rd8m`), and the visualization adds depth to the concept (`1JdE`). The tokenization approach we proposed appears to be *more robust against noise and geometric changes* (`xhZa`). We are also pleased to hear that the paper is *well-written* (`xhZa, 1JdE`). Moreover, the ablation experiments were found to be *sufficient* (`1JdE`), with excellent results that *set new state-of-the-art* (`rd8m, 1hbx`). Prompted by the insightful reviews, we mainly present the following additional experimental results and analyses for the common questions:

- Following the suggestion by Reviewer `rd8m`, we have rethought the evaluation metrics for the BEV map estimation task and additionally explored the boundary quality of the drivable area. The experimental results demonstrate that our method can achieve higher quality boundaries.

- Following the suggestion by Reviewers `rd8m, 1Jde, 1hbx`, we have added a computational complexity comparison using the number of parameters, MACs, and the training time. Please refer to Tab. R1 in the attached PDF. It clearly shows that our approach not only demonstrates strong performance (acknowledged by Reviewer `1hbx`), but also saves much computational cost in comparison to the recent SOTA methods MapPrior and DDP, in both the training and testing phases. In addition, the two-stage training of our approach introduces some additional training overhead in comparison to BEVFusion.

- Following the suggestion by Reviewers `1hbx, 1Jde`, we have added additional visualization comparisons (please refer to Fig. R1 in the attached PDF), including comparisons with MapPrior and DDP. Visualization results demonstrate that our method has significant advantages under unfavorable imaging conditions, such as rainy and night scenes.

- Regarding the concern about dataset split of nuScenes raised by Reviewer `xhZa`, we further supplement our experiments under the surround-view settings to ease the concern with regard to the nuScenes *test* set. We directly tested the off-the-shelf trained models from BEVFusion, DDP and our approach on the nuScenes *test* set to further show the superiority of our approach (although none of the previous works conduct the comparison on the nuScenes *test* set). Our approach consistently outperforms DDP and BEVFusion by large margins.

---

### Decision · Program_Chairs · 2024-09-25

**Decision:**

Accept (poster)

**Comment:**

This paper presents a novel approach for Bird's-Eye-View (BEV) map layout estimation using a generative model inspired by Vector Quantized-Variational AutoEncoder (VQ-VAE). The paper received 4 reviews with mixed paper ratings: 2 Weak Accepts, 1 Borderline Accept and 1 Reject. The reviewers pointed out the following advantages for this paper: novel approach, SOTA performance, clear presentation, and sufficient experimental justification. The authors provided a thorough and thoughtful rebuttal, addressing each of the reviewers' concerns. They also provided additional experiments in the rebuttal, including comparisons with other attention mechanisms, evaluations using different codebook sizes, and computational complexity analysis. Reviewer xhZa gave the Reject suggestion and the main concerns raised by the reviewer are the experimental comparison with SOTA methods and the data split of nuScene dataset in the experiment. The authors addressed these two issues well in the rebuttal, but the reviewer didn't respond to the rebuttal. In consideration of all reviewers suggestions after the rebuttal, the AC suggests to accept the paper.